# Listening to Sounds of Silence for Speech Denoising

**Ruilin Xu[1], Rundi Wu[1], Yuko Ishiwaka[2], Carl Vondrick[1], and Changxi Zheng[1]**

[1]Columbia University, New York, USA
[2]SoftBank Group Corp., Tokyo, Japan

## Abstract

We introduce a deep learning model for speech denoising, a long-standing challenge in audio analysis arising in numerous applications. Our approach is based on a key observation about human speech: there is often a short pause between each sentence or word. In a recorded speech signal, those pauses introduce a series of time periods during which only noise is present. We leverage these incidental *silent intervals* to learn a model for automatic speech denoising given only mono-channel audio. Detected silent intervals over time expose not just pure noise but its time-varying features, allowing the model to learn noise dynamics and suppress it from the speech signal. Experiments on multiple datasets confirm the pivotal role of silent interval detection for speech denoising, and our method outperforms several state-of-the-art denoising methods, including those that accept only audio input (like ours) and those that denoise based on audiovisual input (and hence require more information). We also show that our method enjoys excellent generalization properties, such as denoising spoken languages not seen during training.

## 1 Introduction

Noise is everywhere. When we listen to someone speak, the audio signals we receive are never pure and clean, always contaminated by all kinds of noises—cars passing by, spinning fans in an air conditioner, barking dogs, music from a loudspeaker, and so forth. To a large extent, people in a conversation can effortlessly filter out these noises [42]. In the same vein, numerous applications, ranging from cellular communications to human-robot interaction, rely on speech denoising algorithms as a fundamental building block.

Despite its vital importance, algorithmic speech denoising remains a grand challenge. Provided an input audio signal, speech denoising aims to separate the foreground (speech) signal from its additive background noise. This separation problem is inherently ill-posed. Classic approaches such as spectral subtraction [7, 98, 6, 72, 79] and Wiener filtering [80, 40] conduct audio denoising in the spectral domain, and they are typically restricted to stationary or quasi-stationary noise. In recent years, the advance of deep neural networks has also inspired their use in audio denoising. While outperforming the classic denoising approaches, existing neural-network-based approaches use network structures developed for general audio processing tasks [56, 90, 100] or borrowed from other areas such as computer vision [31, 26, 3, 36, 32] and generative adversarial networks [70, 71]. Nevertheless, beyond reusing well-developed network models as a black box, a fundamental question remains: *What natural structures of speech can we leverage to mold network architectures for better performance on speech denoising?*

### 1.1 Key insight: time distribution of silent intervals

Motivated by this question, we revisit one of the most widely used audio denoising methods in practice, namely the spectral subtraction method [7, 98, 6, 72, 79]. Implemented in many commercial software such as Adobe Audition [39], this classical method requires the user to specify a time interval during which the foreground signal is absent. We call such an interval a *silent interval*. A silent interval is a time window that exposes pure noise. The algorithm then learns from the silent

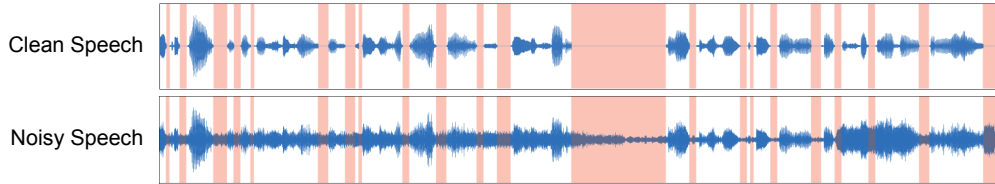

Figure 1: **Silent intervals over time. (top)** A speech signal has many natural pauses. Without any noise, these pauses are exhibited as silent intervals (highlighted in red). **(bottom)** However, most speech signals are contaminated by noise. Even with mild noise, silent intervals become overwhelmed and hard to detect. If robustly detected, silent intervals can help to reveal the noise profile over time.

interval the noise characteristics, which are in turn used to suppress the additive noise of the entire input signal (through subtraction in the spectral domain).

Yet, the spectral subtraction method suffers from two major shortcomings: i) it requires user specification of a silent interval, that is, not fully automatic; and ii) the single silent interval, although undemanding for the user, is insufficient in presence of *nonstationary* noise—for example, a background music. Ubiquitous in daily life, nonstationary noise has time-varying spectral features. The single silent interval reveals the noise spectral features only in that particular time span, thus inadequate for denoising the entire input signal. The success of spectral subtraction pivots on the concept of silent interval; so do its shortcomings.

In this paper, we introduce a deep network for speech denoising that tightly integrates silent intervals, and thereby overcomes many of the limitations of classical approaches. Our goal is not just to identify a single silent interval, but to find as many as possible silent intervals over time. Indeed, silent intervals in speech appear in abundance: psycholinguistic studies have shown that there is almost always a pause after each sentence and even each word in speech [78, 21]. Each pause, however short, provides a silent interval revealing noise characteristics local in time. All together, these silent intervals assemble a time-varying picture of background noise, allowing the neural network to better denoise speech signals, even in presence of nonstationary noise (see Fig. 1).

In short, to interleave neural networks with established denoising pipelines, we propose a network structure consisting of three major components (see Fig. 2): **i)** one dedicated to silent interval detection, **ii)** another that aims to estimate the full noise from those revealed in silent intervals, akin to an inpainting process in computer vision [38], and **iii)** yet another for cleaning up the input signal.

**Summary of results.** Our neural-network-based denoising model accepts a single channel of audio signal and outputs the cleaned-up signal. Unlike some of the recent denoising methods that take as input audiovisual signals (i.e., both audio and video footage), our method can be applied in a wider range of scenarios (e.g., in cellular communication). We conducted extensive experiments, including ablation studies to show the efficacy of our network components and comparisons to several state-of-the-art denoising methods. We also evaluate our method under various signal-to-noise ratios—even under strong noise levels that are not tested against in previous methods. We show that, under a variety of denoising metrics, our method consistently outperforms those methods, including those that accept only audio input (like ours) and those that denoise based on audiovisual input.

The pivotal role of silent intervals for speech denoising is further confirmed by a few key results. Even without supervising on silent interval detection, the ability to detect silent intervals naturally emerges in our network. Moreover, while our model is trained on English speech only, with no additional training it can be readily used to denoise speech in other languages (such as Chinese, Japanese, and Korean). Please refer to the supplementary materials for listening to our denoising results.

## 2 Related Work

**Speech denoising.** Speech denoising [53] is a fundamental problem studied over several decades. Spectral subtraction [7, 98, 6, 72, 79] estimates the clean signal spectrum by subtracting an estimate of the noise spectrum from the noisy speech spectrum. This classic method was followed by spectrogram factorization methods [84]. Wiener filtering [80, 40] derives the enhanced signal by optimizing the mean-square error. Other methods exploit pauses in speech, forming segments of low acoustic energy where noise statistics can be more accurately measured [13, 57, 86, 15, 75, 10, 11]. Statistical model-based methods [14, 34] and subspace algorithms [12, 16] are also studied.

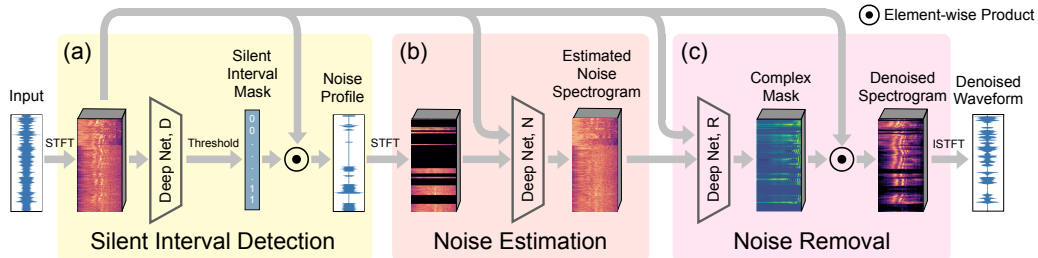

Figure 2: **Our audio denoise network.** Our model has three components: **(a)** one that detects silent intervals over time, and outputs a noise profile observed from detected silent intervals; **(b)** another that estimates the full noise profile, and **(c)** yet another that cleans up the input signal.

Applying neural networks to audio denoising dates back to the 80s [88, 69]. With increased computing power, deep neural networks are often used [104, 106, 105, 47]. Long short-term memory networks (LSTMs) [35] are able to preserve temporal context information of the audio signal [52], leading to strong results [56, 90, 100]. Leveraging generative adversarial networks (GANs) [33], methods such as [70, 71] have adopted GANs into the audio field and have also achieved strong performance.

Audio signal processing methods operate on either the raw waveform or the spectrogram by Short-time Fourier Transform (STFT). Some work directly on waveform [23, 68, 59, 55], and others use Wavenet [91] for speech denoising [74, 76, 30]. Many other methods such as [54, 94, 61, 99, 46, 107, 9] work on audio signal's spectrogram, which contains both magnitude and phase information. There are works discussing how to use the spectrogram to its best potential [93, 67], while one of the disadvantages is that the inverse STFT needs to be applied. Meanwhile, there also exist works [51, 29, 28, 95, 19, 101, 60] investigating how to overcome artifacts from time aliasing.

Speech denoising has also been studied in conjunction with computer vision due to the relations between speech and facial features [8]. Methods such as [31, 26, 3, 36, 32] utilize different network structures to enhance the audio signal to the best of their ability. Adeel et al. [1] even utilize lip-reading to filter out the background noise of a speech.

**Deep learning for other audio processing tasks.** Deep learning is widely used for lip reading, speech recognition, speech separation, and many audio processing or audio-related tasks, with the help of computer vision [64, 66, 5, 4]. Methods such as [50, 17, 65] are able to reconstruct speech from pure facial features. Methods such as [2, 63] take advantage of facial features to improve speech recognition accuracy. Speech separation is one of the areas where computer vision is best leveraged. Methods such as [25, 64, 18, 109] have achieved impressive results, making the previously impossible speech separation from a single audio signal possible. Recently, Zhang et al. [108] proposed a new operation called Harmonic Convolution to help networks distill audio priors, which is shown to even further improve the quality of speech separation.

## 3 Learning Speech Denoising

We present a neural network that harnesses the time distribution of silent intervals for speech denoising. The input to our model is a spectrogram of noisy speech [103, 20, 83], which can be viewed as a 2D image of size $T \times F$ with two channels, where $T$ represents the time length of the signal and $F$ is the number of frequency bins. The two channels store the real and imaginary parts of STFT, respectively. After learning, the model will produce another spectrogram of the same size as the noise suppressed.

We first train our proposed network structure in an end-to-end fashion, with only denoising supervision (Sec. 3.2); and it already outperforms the state-of-the-art methods that we compare against. Furthermore, we incorporate the supervision on silent interval detection (Sec. 3.3) and obtain even better denoising results (see Sec. 4).

### 3.1 Network structure

Classic denoising algorithms work in three general stages: silent interval specification, noise feature estimation, and noise removal. We propose to interweave learning throughout this process: we rethink each stage with the help of a neural network, forming a new speech denoising model. Since we can chain these networks together and estimate gradients, we can efficiently train the model with large-scale audio data. Figure 2 illustrates this model, which we describe below.

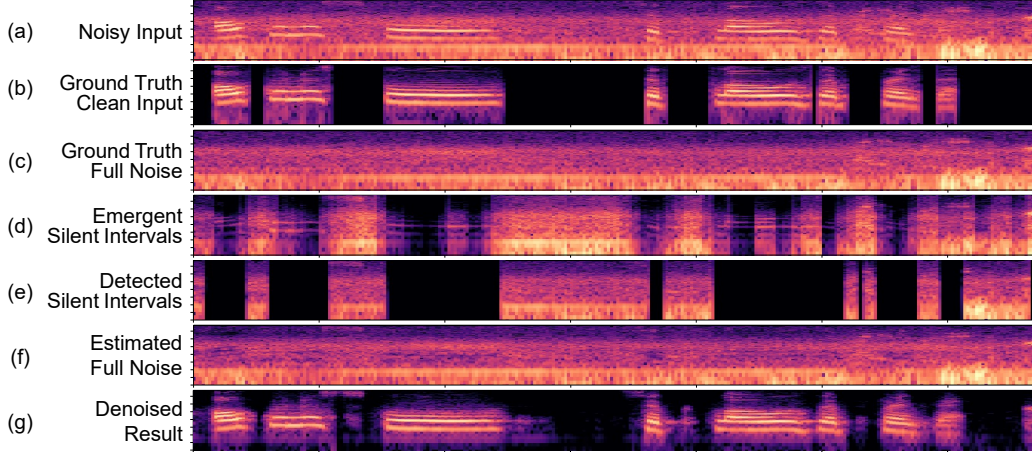

Figure 3: **Example of intermediate and final results. (a)** The spectrogram of a noisy input signal, which is a superposition of a clean speech signal **(b)** and a noise **(c)**. The **black** regions in (b) indicate ground-truth silent intervals. **(d)** The noise exposed by automatically emergent silent intervals, i.e., the output of the silent interval detection component when the entire network is trained without silent interval supervision (recall Sec. 3.2). **(e)** The noise exposed by detected silent intervals, i.e., the output of the silent interval detection component when the network is trained with silent interval supervision (recall Sec. 3.3). **(f)** The estimated noise profile using subfigure (a) and (e) as the input to the noise estimation component. **(g)** The final denoised spectrogram output.

**Silent interval detection.** The first component is dedicated to detecting silent intervals in the input signal. The input to this component is the spectrogram of the input (noisy) signal $x$. The spectrogram $s_x$ is first encoded by a 2D convolutional encoder into a 2D feature map, which is in turn processed by a bidirectional LSTM [35, 81] followed by two fully-connected (FC) layers (see network details in Appendix A). The bidirectional LSTM is suitable for processing time-series features resulting from the spectrogram [58, 41, 73, 18], and the FC layers are applied to the features of each time sample to accommodate variable length input. The output from this network component is a vector $D(s_x)$. Each element of $D(s_x)$ is a scalar in [0,1] (after applying the sigmoid function), indicating a confidence score of a small time segment being silent. We choose each time segment to have $1/30$ second, small enough to capture short speech pauses and large enough to allow robust prediction.

The output vector $D(s_x)$ is then expanded to a longer mask, which we denote as $m(x)$. Each element of this mask indicates the confidence of classifying each sample of the input signal $x$ as pure noise (see Fig. 3-e). With this mask, the noise profile $\tilde{x}$ exposed by silent intervals are estimated by an element-wise product, namely $\tilde{x} := x \odot m(x)$.

**Noise estimation.** The signal $\tilde{x}$ resulted from silent interval detection is noise profile exposed only through a series of time windows (see Fig. 3-e)—but not a complete picture of the noise. However, since the input signal is a superposition of clean speech signal and noise, having a complete noise profile would ease the denoising process, especially in presence of nonstationary noise. Therefore, we also estimate the entire noise profile over time, which we do with a neural network.

Inputs to this component include both the noisy audio signal $x$ and the incomplete noise profile $\tilde{x}$. Both are converted by STFT into spectrograms, denoted as $s_x$ and $s_{\tilde{x}}$, respectively. We view the spectrograms as 2D images. And because the neighboring time-frequency pixels in a spectrogram are often correlated, our goal here is conceptually akin to the image inpainting task in computer vision [38]. To this end, we encode $s_x$ and $s_{\tilde{x}}$ by two separate 2D convolutional encoders into two feature maps. The feature maps are then concatenated in a channel-wise manner and further decoded by a convolutional decoder to estimate the full noise spectrogram, which we denote as $N(s_x, s_{\tilde{x}})$. A result of this step is illustrated in Fig. 3-f.

**Noise removal.** Lastly, we clean up the noise from the input signal $x$. We use a neural network R that takes as input both the input audio spectrogram $s_x$ and the estimated full noise spectrogram $N(s_x, s_{\tilde{x}})$. The two input spectrograms are processed individually by their own 2D convolutional encoders. The two encoded feature maps are then concatenated together before passing to a bidirectional LSTM followed by three fully connected layers (see details in Appendix A). Like other audio enhancement models [18, 92, 96], the output of this component is a vector with two channels which

form the real and imaginary parts of a complex ratio mask $\mathbf{c} := \mathsf{R}\big(\boldsymbol{s_x}, \mathsf{N}(\boldsymbol{s_x}, \boldsymbol{s_{\tilde{x}}})\big)$ in frequency-time domain. In other words, the mask $\mathbf{c}$ has the same (temporal and frequency) dimensions as $\boldsymbol{s_x}$.

In the final step, we compute the denoised spectrogram $\boldsymbol{s_x^*}$ through element-wise multiplication of the input audio spectrogram $\boldsymbol{s_x}$ and the mask $\mathbf{c}$ (i.e., $\boldsymbol{s_x^*} = \boldsymbol{s_x} \odot \mathbf{c}$). Finally, the cleaned-up audio signal is obtained by applying the inverse STFT to $\boldsymbol{s_x^*}$ (see Fig. 3-g).

### 3.2 Loss functions and training

Since a subgradient exists at every step, we are able to train our network in an end-to-end fashion with stochastic gradient descent. We optimize the following loss function:

$$\mathcal{L}_0 = \mathbb{E}_{\boldsymbol{x} \sim p(\boldsymbol{x})}\Big[ \big\| \mathsf{N}(\boldsymbol{s_x}, \boldsymbol{s_{\tilde{x}}}) - \boldsymbol{s_n^*} \big\|_2 + \beta \big\| \boldsymbol{s_x} \odot \mathsf{R}\big(\boldsymbol{s_x}, \mathsf{N}(\boldsymbol{s_x}, \boldsymbol{s_{\tilde{x}}})\big) - \boldsymbol{s_x^*} \big\|_2 \Big], \tag{1}$$

where the notations $\boldsymbol{s_x}$, $\boldsymbol{s_{\tilde{x}}}$, $\mathsf{N}(\cdot, \cdot)$, and $\mathsf{R}(\cdot, \cdot)$ are defined in Sec. 3.1; $\boldsymbol{s_x^*}$ and $\boldsymbol{s_n^*}$ denote the spectrograms of the ground-truth foreground signal and background noise, respectively. The first term penalizes the discrepancy between estimated noise and the ground-truth noise, while the second term accounts for the estimation of foreground signal. These two terms are balanced by the scalar $\beta$ ($\beta = 1.0$ in our experiments).

**Natural emergence of silent intervals.** While producing plausible denoising results (see Sec. 4.4), the end-to-end training process has no supervision on silent interval detection: the loss function (1) only accounts for the recoveries of noise and clean speech signal. But somewhat surprisingly, the ability of detecting silent intervals automatically emerges as the output of the first network component (see Fig. 3-d as an example, which visualizes $\boldsymbol{s_{\tilde{x}}}$). In other words, the network automatically learns to detect silent intervals for speech denoising without this supervision.

### 3.3 Silent interval supervision

As the model is learning to detect silent intervals on its own, we are able to directly supervise silent interval detection to further improve the denoising quality. Our first attempt was to add a term in (1) that penalizes the discrepancy between detected silent intervals and their ground truth. But our experiments show that this is not effective (see Sec. 4.4). Instead, we train our network in two sequential steps.

First, we train the silent interval detection component through the following loss function:

$$\mathcal{L}_1 = \mathbb{E}_{\boldsymbol{x} \sim p(\boldsymbol{x})}\Big[ \ell_{\mathrm{BCE}}\big(\boldsymbol{m}(\boldsymbol{x}), \boldsymbol{m_x^*}\big) \Big], \tag{2}$$

where $\ell_{\mathrm{BCE}}(\cdot, \cdot)$ is the binary cross entropy loss, $\boldsymbol{m}(\boldsymbol{x})$ is the mask resulted from silent interval detection component, and $\boldsymbol{m_x^*}$ is the ground-truth label of each signal sample being silent or not—the way of constructing $\boldsymbol{m_x^*}$ and the training dataset will be described in Sec. 4.1.

Next, we train the noise estimation and removal components through the loss function (1). This step starts by neglecting the silent detection component. In the loss function (1), instead of using $\boldsymbol{s_{\tilde{x}}}$, the noise spectrogram exposed by the estimated silent intervals, we use the noise spectrogram exposed by the ground-truth silent intervals (i.e., the STFT of $\boldsymbol{x} \odot \boldsymbol{m_x^*}$). After training using such a loss function, we fine-tune the network components by incorporating the already trained silent interval detection component. With the silent interval detection component fixed, this fine-tuning step optimizes the original loss function (1) and thereby updates the weights of the noise estimation and removal components.

## 4 Experiments

This section presents the major evaluations of our method, comparisons to several baselines and prior works, and ablation studies. We also refer the reader to the supplementary materials (including a supplemental document and audio effects organized on an off-line webpage) for the full description of our network structure, implementation details, additional evaluations, as well as audio examples.

### 4.1 Experiment setup

**Dataset construction.** To construct training and testing data, we leverage publicly available audio datasets. We obtain clean speech signals using AVSPEECH [18], from which we randomly choose 2448 videos (4.5 hours of total length) and extract their speech audio channels. Among them, we use 2214 videos for training and 234 videos for testing, so the training and testing speeches are fully

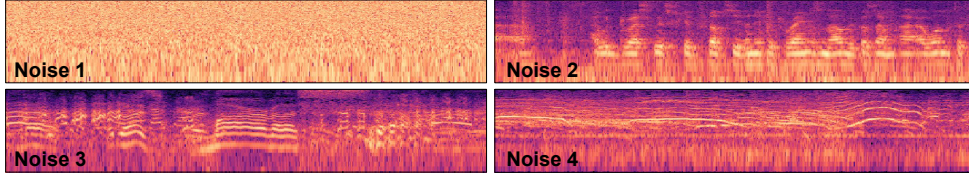

Figure 4: **Noise gallery.** We show four examples of noise from the noise datasets. Noise 1) is a stationary (white) noise, and the other three are not. Noise 2) is a monologue in a meeting. Noise 3) is party noise from people speaking and laughing with background noise. Noise 4) is street noise from people shouting and screaming with additional traffic noise such as vehicles driving and honking.

separate. All these speech videos are in English, selected on purpose: as we show in supplementary materials, our model trained on this dataset can readily denoise speeches in other languages.

We use two datasets, `DEMAND` [89] and Google's `AudioSet` [27], as background noise. Both consist of environmental noise, transportation noise, music, and many other types of noises. `DEMAND` has been used in previous denoising works (e.g., [70, 30, 90]). Yet `AudioSet` is much larger and more diverse than `DEMAND`, thus more challenging when used as noise. Figure 4 shows some noise examples. Our evaluations are conducted on both datasets, separately.

Due to the linearity of acoustic wave propagation, we can superimpose clean speech signals with noise to synthesize noisy input signals (similar to previous works [70, 30, 90]). When synthesizing a noisy input signal, we randomly choose a signal-to-noise ratio (SNR) from seven discrete values: -10dB, -7dB, -3dB, 0dB, 3dB, 7dB, and 10dB; and by mixing the foreground speech with properly scaled noise, we produce a noisy signal with the chosen SNR. For example, a -10dB SNR means that the power of noise is ten times the speech (see Fig. S1 in appendix). The SNR range in our evaluations (i.e., [-10dB, 10dB]) is significantly larger than those tested in previous works.

To supervise our silent interval detection (recall Sec. 3.3), we need ground-truth labels of silent intervals. To this end, we divide each clean speech signal into time segments, each of which lasts $1/30$ seconds. We label a time segment as silent when the total acoustic energy in that segment is below a threshold. Since the speech is clean, this automatic labeling process is robust.

*Remarks on creating our own datasets.* Unlike many previous models, which are trained using existing datasets such as Valentini's VoiceBank-DEMAND [90], we choose to create our own datasets because of two reasons. First, Valentini's dataset has a noise SNR level in [0dB, 15dB], much narrower than what we encounter in real-world recordings. Secondly, although Valentini's dataset provides several kinds of environmental noise, it lacks the richness of other types of structured noise such as music, making it less ideal for denoising real-world recordings (see discussion in Sec. 4.6).

**Method comparison.** We compare our method with several existing methods that are also designed for speech denoising, including both the classic approaches and recently proposed learning-based methods. We refer to these methods as follows: **i)** `Ours`, our model trained with silent interval supervision (recall Sec. 3.3); **ii)** `Baseline-thres`, a baseline method that uses acoustic energy threshold to label silent intervals (the same as our automatic labeling approach in Sec. 4.1 but applied on noisy input signals), and then uses our trained noise estimation and removal networks for speech denoising. **iii)** `Ours-GTSI`, another reference method that uses our trained noise estimation and removal networks, but hypothetically uses the ground-truth silent intervals; **iv)** `Spectral Gating`, the classic speech denoising algorithm based on spectral subtraction [79]; **v)** `Adobe Audition` [39], one of the most widely used professional audio processing software, and we use its machine-learning-based noise reduction feature, provided in the latest Adobe Audition CC 2020, with default parameters to batch process all our test data; **vi)** `SEGAN` [70], one of the state-of-the-art audio-only speech enhancement methods based on generative adversarial networks. **vii)** `DFL` [30], a recently proposed speech denoising method based on a loss function over deep network features; [1] **viii)** `VSE` [26], a learning-based method that takes both video and audio as input, and leverages both audio signal and mouth motions (from video footage) for speech denoising. We could not compare with another audiovisual method [18] because no source code or executable is made publicly available.

For fair comparisons, we train all the methods (except `Spectral Gating` which is not learning-based and `Adobe Audition` which is commercially shipped as a black box) using the same datasets.

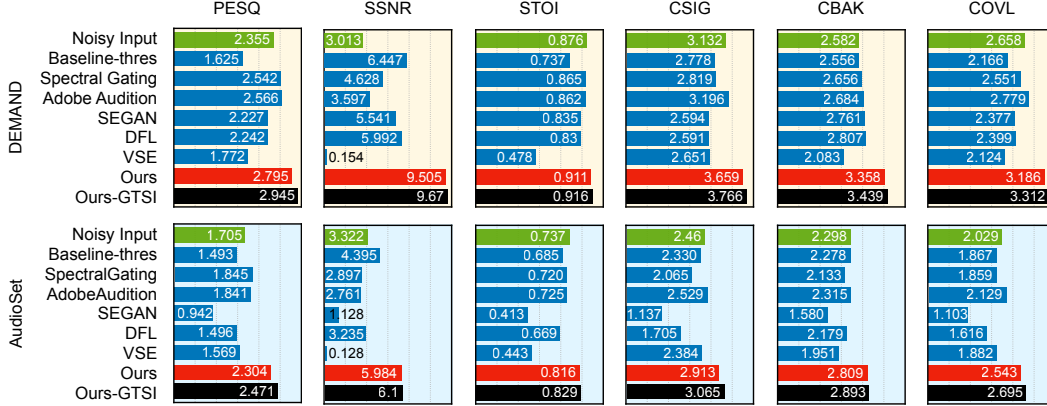

Figure 5: **Quantitative comparisons.** We measure denoising quality under six metrics (corresponding to columns). The comparisons are conducted using noise from `DEMAND` and `AudioSet` separately. `Ours-GTSI` (in black) uses ground-truth silent intervals. Although not a practical approach, it serves as an upper-bound reference of all methods. Meanwhile, the green bar in each plot indicates the metric score of the noisy input without any processing.

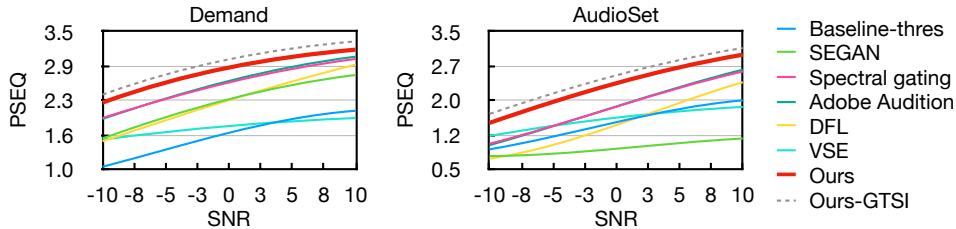

Figure 6: **Denoise quality w.r.t. input SNRs.** Denoise results measured in PESQ for each method w.r.t different input SNRs. Results measured in other metrics are shown in Fig. S2 in Appendix.

For `SEGAN`, `DFL`, and `VSE`, we use their source codes published by the authors. The audiovisual denoising method `VSE` also requires video footage, which is available in `AVSPEECH`.

## 4.2 Evaluation on speech denoising

**Metrics.** Due to the perceptual nature of audio processing tasks, there is no widely accepted single metric for quantitative evaluation and comparisons. We therefore evaluate our method under six different metrics, all of which have been frequently used for evaluating audio processing quality. Namely, these metrics are: **i)** Perceptual Evaluation of Speech Quality (PESQ) [77], **ii)** Segmental Signal-to-Noise Ratio (SSNR) [82], **iii)** Short-Time Objective Intelligibility (STOI) [87], **iv)** Mean opinion score (MOS) predictor of signal distortion (CSIG) [37], **v)** MOS predictor of background-noise intrusiveness (CBAK) [37], and **vi)** MOS predictor of overall signal quality (COVL) [37].

**Results.** We train two separate models using `DEMAND` and `AudioSet` noise datasets respectively, and compare them with other models trained with the same datasets. We evaluate the average metric values and report them in Fig. 5. Under all metrics, our method consistently outperforms others.

We breakdown the performance of each method with respect to SNR levels from -10dB to 10dB on both noise datasets. The results are reported in Fig. 6 for PESQ (see Fig. S2 in the appendix for all metrics). In the previous works that we compare to, no results under those low SNR levels (at < 0 dBs) are reported. Nevertheless, across all input SNR levels, our method performs the best, showing that our approach is fairly robust to both light and extreme noise.

From Fig. 6, it is worth noting that `Ours-GTSI` method performs even better. Recall that this is our model but provided with ground-truth silent intervals. While not practical (due to the need of ground-truth silent intervals), `Ours-GTSI` confirms the importance of silent intervals for denoising: a high-quality silent interval detection helps to improve speech denoising quality.

## 4.3 Evaluation on silent interval detection

Due to the importance of silent intervals for speech denoising, we also evaluate the quality of our silent interval detection, in comparison to two alternatives, the baseline `Baseline-thres` and a

Table 1: **Results of silent interval detection.** The metrics are measured using our test signals that have SNRs from -10dB to 10dB. Definitions of these metrics are summarized in Appendix C.1.

| Noise Dataset | Method | Precision | Recall | F1 | Accuracy |
|---|---|---|---|---|---|
| DEMAND | Baseline-thres | 0.533 | 0.718 | 0.612 | 0.706 |
| | VAD | 0.797 | 0.432 | 0.558 | 0.783 |
| | Ours | **0.876** | **0.866** | **0.869** | **0.918** |
| Audioset | Baseline-thres | 0.536 | 0.731 | 0.618 | 0.708 |
| | VAD | 0.736 | 0.227 | 0.338 | 0.728 |
| | Ours | **0.794** | **0.822** | **0.807** | **0.873** |

Table 2: **Ablation studies.** We alter network components and training loss, and evaluate the denoising quality under various metrics. Our proposed approach performs the best.

| Noise Dataset | Method | PESQ | SSNR | STOI | CSIG | CBAK | COVL |
|---|---|---|---|---|---|---|---|
| DEMAND | Ours w/o SID comp | 2.689 | 9.080 | 0.904 | 3.615 | 3.285 | 3.112 |
| | Ours w/o NR comp | 2.476 | 0.234 | 0.747 | 3.015 | 2.410 | 2.637 |
| | Ours w/o SID loss | 2.794 | 6.478 | 0.903 | 3.466 | 3.147 | 3.079 |
| | Ours w/o NE loss | 2.601 | 9.070 | 0.896 | 3.531 | 3.237 | 3.027 |
| | Ours Joint loss | 2.774 | 6.042 | 0.895 | 3.453 | 3.121 | 3.068 |
| | Ours | **2.795** | **9.505** | **0.911** | **3.659** | **3.358** | **3.186** |
| Audioset | Ours w/o SID comp | 2.190 | 5.574 | 0.802 | 2.851 | 2.719 | 2.454 |
| | Ours w/o NR comp | 1.803 | 0.191 | 0.623 | 2.301 | 2.070 | 1.977 |
| | Ours w/o SID loss | **2.325** | 4.957 | 0.814 | 2.814 | 2.746 | 2.503 |
| | Ours w/o NE loss | 2.061 | 5.690 | 0.789 | 2.766 | 2.671 | 2.362 |
| | Ours Joint loss | 2.305 | 4.612 | 0.807 | 2.774 | 2.721 | 2.474 |
| | Ours | 2.304 | **5.984** | **0.816** | **2.913** | **2.809** | **2.543** |

Voice Activity Detector (VAD) [22]. The former is described above, while the latter classifies each time window of an audio signal as having human voice or not [48, 49]. We use an off-the-shelf VAD [102], which is developed by Google's WebRTC project and reported as one of the best available. Typically, VAD is designed to work with low-noise signals. Its inclusion here here is merely to provide an alternative approach that can detect silent intervals in more ideal situations.

We evaluate these methods using four standard statistic metrics: the precision, recall, F1 score, and accuracy. We follow the standard definitions of these metrics, which are summarized in Appendix C.1. These metrics are based on the definition of positive/negative conditions. Here, the positive condition indicates a time segment being labeled as a silent segment, and the negative condition indicates a non-silent label. Thus, the higher the metric values are, the better the detection approach.

Table 1 shows that, under all metrics, our method is consistently better than the alternatives. Between VAD and Baseline-thres, VAD has higher precision and lower recall, meaning that VAD is overly conservative and Baseline-thres is overly aggressive when detecting silent intervals (see Fig. S3 in Appendix C.2). Our method reaches better balance and thus detects silent intervals more accurately.

## 4.4 Ablation studies

In addition, we perform a series of ablation studies to understand the efficacy of individual network components and loss terms (see Appendix D.1 for more details). In Table 2, "Ours w/o SID loss" refers to the training method presented in Sec. 3.2 (i.e., without silent interval supervision). "Ours Joint loss" refers to the end-to-end training approach that optimizes the loss function (1) with the additional term (2). And "Ours w/o NE loss" uses our two-step training (in Sec. 3.3) but without the loss term on noise estimation—that is, without the first term in (1). In comparison to these alternative training approaches, our two-step training with silent interval supervision (referred to as "Ours") performs the best. We also note that "Ours w/o SID loss"—i.e., without supervision on silent interval detection—already outperforms the methods we compared to in Fig. 5, and "Ours" further improves the denoising quality. This shows the efficacy of our proposed training approach.

We also experimented with two variants of our network structure. The first one, referred to as "Ours w/o SID comp", turns off silent interval detection: the silent interval detection component always outputs a vector with all zeros. The second, referred as "Ours w/o NR comp", uses a simple spectral

Table 3: Comparisons on VoiceBank-DEMAND corpus.

| Method | PESQ | CSIG | CBAK | COVL | STOI |
|---|---|---|---|---|---|
| Noisy Input | 1.97 | 3.35 | 2.44 | 2.63 | 0.91 |
| WaveNet [91] | – | 3.62 | 3.24 | 2.98 | – |
| SEGAN [70] | 2.16 | 3.48 | 2.94 | 2.80 | 0.93 |
| DFL [30] | 2.51 | 3.79 | 3.27 | 3.14 | – |
| MMSE-GAN [85] | 2.53 | 3.80 | 3.12 | 3.14 | 0.93 |
| MetricGAN [24] | 2.86 | 3.99 | 3.18 | 3.42 | – |
| SDR-PESQ [43] | 3.01 | 4.09 | 3.54 | 3.55 | – |
| T-GSA [44] | 3.06 | 4.18 | 3.59 | 3.62 | – |
| Self-adapt. DNN [45] | 2.99 | 4.15 | 3.42 | 3.57 | – |
| RDL-Net [62] | 3.02 | 4.38 | 3.43 | 3.72 | 0.94 |
| Ours | 3.16 | 3.96 | 3.54 | 3.53 | 0.98 |

subtraction to replace our noise removal component. Table 2 shows that, under all the tested metrics, both variants perform worse than our method, suggesting our proposed network structure is effective.

Furthermore, we studied to what extent the accuracy of silent interval detection affects the speech denoising quality. We show that as the silent interval detection becomes less accurate, the denoising quality degrades. Presented in details in Appendix D.2, these experiments reinforce our intuition that silent intervals are instructive for speech denoising tasks.

### 4.5 Comparison with state-of-the-art benchmark

Many state-of-the-art denoising methods, including MMSE-GAN [85], Metric-GAN [24], SDR-PESQ [43], T-GSA [44], Self-adaptation DNN [45], and RDL-Net [62], are all evaluated on Valentini's VoiceBank-DEMAND [90]. We therefore compare ours with those methods on the same dataset. We note that DEMAND consists of audios with SNR in [0dB, 15dB]. Its SNR range is much narrower than what our method (and our training datasets) aims for (e.g., input signals with -10dB SNR). Nevertheless, trained and tested under the same setting, our method is highly competitive to the best of those methods under every metric, as shown in Table 3. The metric scores therein for other methods are numbers reported in their original papers.

### 4.6 Tests on real-world data

We also test our method against real-world data. Quantitative evaluation on real-world data, however, is not easy because the evaluation of nearly all metrics requires the corresponding ground-truth clean signal, which is not available in real-world scenario. Instead, we collected a good number of real-world audios, either by recording in daily environments or by downloading online (e.g., from YouTube). These real-world audios cover diverse scenarios: in a driving car, a café, a park, on the street, in multiple languages (Chinese, Japanese, Korean, German, French, etc.), with different genders and accents, and even with singing songs. None of these recordings is cherry picked. We refer the reader to our project website for the denoising results of all the collected real-world recordings, and for the comparison of our method with other state-of-the-art methods under real-world settings.

Furthermore, we use real-world data to test our model trained with different datasets, including our own dataset (recall Sec. 4.1) and the existing DEMAND [90]. We show that the network model trained by our own dataset leads to much better noise reduction (see details in Appendix E.2). This suggests that our dataset allows the denoising model to better generalize to many real-world scenarios.

## 5 Conclusion

Speech denoising has been a long-standing challenge. We present a new network structure that leverages the abundance of silent intervals in speech. Even without silent interval supervision, our network is able to denoise speech signals plausibly, and meanwhile, the ability to detect silent intervals automatically emerges. We reinforce this ability. Our explicit supervision on silent intervals enables the network to detect them more accurately, thereby further improving the performance of speech denoising. As a result, under a variety of denoising metrics, our method consistently outperforms several state-of-the-art audio denoising models.

**Acknowledgments.** This work was supported in part by the National Science Foundation (1717178, 1816041, 1910839, 1925157) and SoftBank Group.

## Broader Impact

High-quality speech denoising is desired in a myriad of applications: human-robot interaction, cellular communications, hearing aids, teleconferencing, music recording, filmmaking, news reporting, and surveillance systems to name a few. Therefore, we expect our proposed denoising method—be it a system used in practice or a foundation for future technology—to find impact in these applications.

In our experiments, we train our model using English speech only, to demonstrate its generalization property—the ability of denoising spoken languages beyond English. Our demonstration of denoising Japanese, Chinese, and Korean speeches is intentional: they are linguistically and phonologically distant from English (in contrast to other English "siblings" such as German and Dutch). Still, our model may bias in favour of spoken languages and cultures that are closer to English or that have frequent pauses to reveal silent intervals. Deeper understanding of this potential bias requires future studies in tandem with linguistic and sociocultural insights.

Lastly, it is natural to extend our model for denoising audio signals in general or even signals beyond audio (such as Gravitational wave denoising [97]). If successful, our model can bring in even broader impacts. Pursuing this extension, however, requires a judicious definition of "silent intervals". After all, the notion of "noise" in a general context of signal processing depends on specific applications: noise in one application may be another's signals. To train a neural network that exploits a general notion of silent intervals, prudence must be taken to avoid biasing toward certain types of noise.

## Footnotes

[1] This recent method is designed for high-noise-level input, trained in an end-to-end fashion, and as their paper states, is "particularly pronounced for the hardest data with the most intrusive background noise".

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
