[Supplementary Material · Supplementary Document.pdf]

# Supplementary Document
# Listening to Sounds of Silence for Speech Denoising

## A   Network Structure and Training Details

We now present the details of our network structure and training configurations.

**The silent interval detection component** of our model is composed of 2D convolutional layers, a bidirectional LSTM, and two FC layers. The parameters of the convolutional layers are shown in Table S1. Each convolutional layer is followed by a batch normalization layer with a ReLU activation function. The hidden size of bidirectional LSTM is 100. The two FC layers, interleaved with a ReLU activation function, have hidden size of 100 and 1, respectively.

Table S1: Convolutional layers in the silent interval detection component.

|  | conv1 | conv2 | conv3 | conv4 | conv5 | conv6 | conv7 | conv8 | conv9 | conv10 | conv11 | conv12 |
|---|---|---|---|---|---|---|---|---|---|---|---|---|
| Num Filters | 48 | 48 | 48 | 48 | 48 | 48 | 48 | 48 | 48 | 48 | 48 | 8 |
| Filter Size | (1,7) | (7,1) | (5,5) | (5,5) | (5,5) | (5,5) | (5,5) | (5,5) | (5,5) | (5,5) | (5,5) | (1,1) |
| Dilation | (1,1) | (1,1) | (1,1) | (2,1) | (4,1) | (8,1) | (16,1) | (32,1) | (1,1) | (2,2) | (4,4) | (1,1) |
| Stride | 1 | 1 | 1 | 1 | 1 | 1 | 1 | 1 | 1 | 1 | 1 | 1 |

**The noise estimation component** of our model is fully convolutional, consisting of two encoders and one decoder. The two encoders process the noisy signal and the incomplete noise profile, respectively; they have the same architecture (shown in Table S2) but different weights. The two feature maps resulted from the two encoders are concatenated in a channel-wise manner before feeding into the decoder. In Table S2, every layer, except the last one, is followed by a batch normalization layer together with a ReLU activation function. In addition, there is skip connections between the 2nd and 14-th layer and between the 4-th and 12-th layer.

Table S2: Architecture of noise estimation component. 'C' indicates a convolutional layer, and 'TC' indicates a transposed convolutional layer.

|  | Encoder | | | Decoder | | | | | | | | | | | | |
|---|---|---|---|---|---|---|---|---|---|---|---|---|---|---|---|---|
| ID | 1 | 2 | 3 | 4 | 5 | 6 | 7 | 8 | 9 | 10 | 11 | 12 | 13 | 14 | 15 | 16 |
| Layer Type | C | C | C | C | C | C | C | C | C | C | C | TC | C | TC | C | C |
| Num Filters | 64 | 128 | 128 | 256 | 256 | 256 | 256 | 256 | 256 | 256 | 256 | 128 | 128 | 64 | 64 | 2 |
| Filter Size | 5 | 5 | 5 | 3 | 3 | 3 | 3 | 3 | 3 | 3 | 3 | 3 | 3 | 3 | 3 | 3 |
| Dilation | 1 | 1 | 1 | 1 | 1 | 2 | 4 | 8 | 16 | 1 | 1 | 1 | 1 | 1 | 1 | 1 |
| Stride | 1 | 2 | 1 | 2 | 1 | 1 | 1 | 1 | 1 | 1 | 1 | 2 | 1 | 2 | 1 | 1 |

**The noise removal component** of our model is composed of two 2D convolutional encoders, a bidirectional LSTM, and three FC layers. The two convolutional encoders take as input the input audio spectrogram $s_x$ and the estimated full noise spectrogram $\mathsf{N}(s_x, s_{\tilde{x}})$, respectively. The first encoder has the network architecture listed in Table S3, and the second has the same architecture but with half of the number of filters at each convolutional layer. Moreover, the bidirectional LSTM has the hidden size of 200, and the three FC layers have the hidden size of 600, 600, and $2F$, respectively, where $F$ is the number of frequency bins in the spectrogram. In terms of the activation function, ReLU is used after each layer except the last layer, which uses sigmoid.

Table S3: Convolutional encoder for the noise removal component of our model. Each convolutional layer is followed by a batch normalization layer with ReLU as the activation function.

|  | C1 | C2 | C3 | C4 | C5 | C6 | C7 | C8 | C9 | C10 | C11 | C12 | C13 | C14 | C15 |
|---|---|---|---|---|---|---|---|---|---|---|---|---|---|---|---|
| Num Filters | 96 | 96 | 96 | 96 | 96 | 96 | 96 | 96 | 96 | 96 | 96 | 96 | 96 | 96 | 8 |
| Filter Size | (1,7) | (7,1) | (5,5) | (5,5) | (5,5) | (5,5) | (5,5) | (5,5) | (5,5) | (5,5) | (5,5) | (5,5) | (5,5) | (5,5) | (1,1) |
| Dilation | (1,1) | (1,1) | (1,1) | (2,1) | (4,1) | (8,1) | (16,1) | (32,1) | (1,1) | (2,2) | (4,4) | (8,8) | (16,16) | (32,32) | (1,1) |
| Stride | 1 | 1 | 1 | 1 | 1 | 1 | 1 | 1 | 1 | 1 | 1 | 1 | 1 | 1 | 1 |

Figure S1: **Constructed noisy audio based on different SNR levels.** The first row shows the waveform of the ground truth clean input.

**Training details.**  We use PyTorch platform to implement our speech denoising model, which is then trained with the Adam optimizer. In our end-to-end training without silent interval supervision (referred to as "`Ours w/o SID loss`" in Sec. 4; also recall Sec. 3.2), we run the Adam optimizer for 50 epochs with a batch size of 20 and a learning rate of 0.001. When the silent interval supervision is incorporated (recall Sec. 3.3), we first train the silent interval detection component with the following setup: run the Adam optimizer for 100 epochs with a batch size of 15 and a learning rate of 0.001. Afterwards, we train the noise estimation and removal components using the same setup as the end-to-end training of "`Ours w/o SID loss`".

## B   Data Processing Details

Our model is designed to take as input a mono-channel audio clip of an arbitrary length. However, when constructing the training dataset, we set each audio clip in the training dataset to have the same 2-second length, to enable batching at training time. To this end, we split each original audio clip from `AVSPEECH`, `DEMAND`, and `AudioSet` into 2-second long clips. All audio clips are then downsampled at 16kHz before converting into spectrograms using STFT. To perform STFT, the Fast Fourier Transform (FFT) size is set to 510, the Hann window size is set to 28ms, and the hop length is set to 11ms. As a result, each 2-second clip yields a (complex-valued) spectrogram with a resolution $256 \times 178$, where 256 is the number of frequency bins, and 178 is the temporal resolution. At inference time, our model can still accept audio clips with arbitrary length.

Both our clean speech dataset and noise datasets are first split into training and test sets, so that no audio clips in training and testing are from the same original audio source—they are fully separate.

To supervise our silent interval detection, we label the clean audio signals in the following way. We first normalize each audio clip so that its magnitude is in the range [-1,1], that is, ensuring the largest waveform magnitude at -1 or 1. Then, the clean audio clip is divided into segments of length $1/30$ seconds. We label a time segment as a "silent" segment (i.e., label 0) if its average waveform energy in that segment is below 0.08. Otherwise, it is labeled as a "non-silent" segment (i.e., label 1).

Figure S2: **Denoise quality under different input SNRs.** Here we expand Fig. 6 of the main text, including the evaluations under all six metrics described in Sec. 4.2.

## C  Evaluation on Silent Interval Detection

### C.1  Metrics

We now provide the details of the metrics used for evaluating our silent interval detection (i.e., results in Table 1 of the main text). Detecting silent intervals is a binary classification task, one that classifies

every time segment as being silent (i.e., a positive condition) or not (i.e., a negative condition). Recall that the confusion matrix in a binary classification task is as follows:

Table S4: Confusion matrix

|  |  | Actual | |
|---|---|---|---|
|  |  | Positive | Negative |
| Predicted | Positive | True Positive (TP) | False Positive (FP) |
|  | Negative | False Negative (FN) | True Negative (TN) |

In our case, we have the following conditions:

- A true positive (TP) sample is a correctly predicted silent segment.
- A true negative (TN) sample is a correctly predicted non-silent segment.
- A false positive (FP) sample is a non-silent segment predicted as silent.
- A false negative (FN) sample is a silent segment predicted as non-silent.

The four metrics used in Table 1 follow the standard definitions in statistics, which we review here:

$$
\begin{aligned}
\text{precision} &= \frac{N_{\mathsf{TP}}}{N_{\mathsf{TP}} + N_{\mathsf{FP}}}, \\
\text{recall} &= \frac{N_{\mathsf{TP}}}{N_{\mathsf{TP}} + N_{\mathsf{FN}}}, \\
\text{F1} &= 2 \cdot \frac{\text{precision} \cdot \text{recall}}{\text{precision} + \text{recall}}, \ \text{and} \\
\text{accuracy} &= \frac{N_{\mathsf{TP}} + N_{\mathsf{TN}}}{N_{\mathsf{TP}} + N_{\mathsf{TN}} + N_{\mathsf{FP}} + N_{\mathsf{FN}}},
\end{aligned}
\tag{S1}
$$

where $N_{\mathsf{TP}}$, $N_{\mathsf{TN}}$, $N_{\mathsf{FP}}$, and $N_{\mathsf{FN}}$ indicate the numbers of true positive, true negative, false positive, and false negative predictions among all tests. Intuitively, *recall* indicates the ability of correctly finding all true silent intervals, *precision* measures how much proportion of the labeled silent intervals are truly silent. *F1* score takes both precision and recall into account, and produces their harmonic mean. And *accuracy* is the ratio of correct predictions among all predictions.

### C.2  An Example of Silent Interval Detection

In Fig. S3, we present an example of silent interval detection results in comparison to two alternative methods. The two alternatives, described in Sec. 4.3, are referred to as `Baseline-thres` and `VAD`, respectively. Figure S3 echos the quantitative results in Table 1: `VAD` tends to be overly conservative, even in the presence of mild noise; and many silent intervals are ignored. On the other hand, `Baseline-thres` tends to be too aggressive; it produces many false intervals. In contrast, our silent interval detection maintains a better balance, and thus predicts more accurately.

Figure S3: **An example of silent interval detection results.** Provided an input signal whose SNR is 0dB (top-left), we show the silent intervals (in red) detected by three approaches: our method, `Baseline-thres`, and `VAD`. We also show ground-truth silent intervals in top-left.

# D   Ablation Studies and Analysis

## D.1   Details of Ablation Studies

In Sec. 4.4 and Table 2, the ablation studies are set up in the following way.

- "Ours" refers to our proposed network structure and training method that incorporates silent interval supervision (recall Sec. 3.3). Details are described in Appendix A.

- "Ours w/o SID loss" refers to our proposed network structure but optimized by the training method in Sec. 3.2 (i.e. an end-to-end training without silent interval supervision). This ablation study is to confirm that silent interval supervision indeed helps to improve the denoising quality.

- "Ours Joint loss" refers to our proposed network structure optimized by the end-to-end training approach that optimizes the loss function (1) with the additional term (2). In this end-to-end training, silent interval detection is also supervised through the loss function. This ablation study is to confirm that our two-step training (Sec. 3.3) is more effective.

- "Ours w/o NE loss" uses our two-step training (in Sec. 3.3) but without the loss term on noise estimation—that is, without the first term in (1). This ablation study is to examine the necessity of the loss term on noise estimation for better denoising quality.

- "Ours w/o SID comp" turns off silent interval detection: the silent interval detection component always outputs a vector with all zeros. As a result, the input noise profile to the noise estimation component N is made precisely the same as the original noisy signal. This ablation study is to examine the effect of silent intervals for speech denoising.

- "Ours w/o NR comp" uses a simple spectral subtraction to replace our noise removal component; the other components remain unchanged. This ablation studey is to examine the efficacy of our noise removal component.

## D.2   The Influence of Silent Interval Detection on Denoising Quality

A key insight of our neural-network-based denoising model is the leverage of silent interval distribution over time. The experiments above have confirmed the efficacy of our silent interval detection for better speech denoising. We now report additional experiments, aiming to gain some empirical understanding of how the quality of silent interval prediction would affect speech denoising quality.

Table S5: Results on how silent interval detection quality affects the speech denoising quality.

| Shift | No Shift | 1/30 | 1/10 | 1/5 | 1/2 | Shrink | No Shrink | 20% | 40% | 60% | 80% |
|-------|----------|------|------|-----|-----|--------|-----------|-----|-----|-----|-----|
| PESQ | **2.471** | 1.932 | 1.317 | 1.169 | 1.094 | PESQ | **2.471** | 2.361 | 2.333 | 2.283 | 2.249 |

(a) Effect of shifting silent intervals.   (b) Effect of shrinking silent intervals.

First, starting with ground-truth silent intervals, we shift them on the time axis by $1/30$, $1/10$, $1/5$, and $1/2$ seconds. As the shifted time amount increases, more time segments become incorrectly labeled: both the numbers of false positive labels (i.e., non-silent time segments labeled silent) and false negative labels (i.e., silent time segments are labeled non-silent) increase. After each shift, we feed the silent interval labels to our noise estimation and removal components and measure the denoising quality under the PESQ score.

In the second experiment, we again start with ground-truth silent intervals; but instead of shifting them, we shrink each silent interval toward its center by $20\%$, $40\%$, $60\%$, and $80\%$. As the silent intervals become more shrunken, fewer time segments are labeled as silent. In other words, only the number of false negative predictions increases. Similar to the previous experiment, after each shrink, we use the silent interval labels in our speech denoising pipeline, and meausure the PESQ score.

The results of both experiments are reported in Table S5. As we shrink the silent intervals, the denoising quality drops gently. In contrast, even a small amount of shift causes a clear drop of denoising quality. These results suggest that in comparison to false negative predictions, false positive predictions affect the denoising quality more negatively. On the one hand, reasonably conservative predictions may leave certain silent time segments undetected (i.e., introducing some false negative

labels), but the detected silent intervals indeed reveal the noise profile. On the other hand, even a small amount of false positive predictions causes certain non-silent time segments to be treated as silent segments, and thus the observed noise profile through the detected silent intervals would be tainted by foreground signals.

# E   Evaluation of Model Performance

## E.1   Generalization Across Datasets

To evaluate the generalization ability of our model, we performed *three* cross-dataset tests reported in Table S6. The experiments are set up in the following way.

- "Test **i**": We train our model on our own AVSPEECH+Audioset (AA) dataset but evaluate on Valentini's VoiceBank-DEMAND (VD) testset. The result is shown in the first row of the "Test **i**" section in Table S6. In comparison, the second row shows the result of training on VD and testing on VD.

- "Test **ii**": We train our model on our own AA dataset but evaluate on our second AVSPEECH+DEMAND (AD) testset. The result is shown in the first row of the "Test **ii**" section in Table S6. In comparison, the second row shows the result of training on AD and testing on AD.

- "Test **iii**": We train our model on our own AD dataset but evaluate on AA testset. The result is shown in the first row of the "Test **iii**" section of the table. In comparison, the second row shows the result of training on AA and testing on AA.

The small degradation in each cross-dataset test demonstrates the great generalization ability of our method. We could not directly compare the generalization ability of our model with existing methods, as no previous work reported cross-dataset evaluation results.

Table S6: Generalization across datasets.

| Test | Trainset | Testset | PESQ | CSIG | CBAK | COVL | STOI |
|---|---|---|---|---|---|---|---|
| Test **i** | AA | VD | 3.00 | 3.78 | 3.08 | 3.34 | 0.98 |
|  | VD | VD | 3.16 | 3.96 | 3.54 | 3.53 | 0.98 |
| Test **ii** | AA | AD | 2.65 | 3.48 | 3.21 | 3.01 | 0.90 |
|  | AD | AD | 2.80 | 3.66 | 3.36 | 3.17 | 0.91 |
| Test **iii** | AD | AA | 2.12 | 2.71 | 2.65 | 2.34 | 0.79 |
|  | AA | AA | 2.30 | 2.91 | 2.81 | 2.54 | 0.82 |

## E.2   Generalization on Real-world Data

We conduct experiments to understand the extent to which the model trained with different datasets can generalize to real-world data. We train two versions of our model using our AVSPEECH+Audioset dataset and Valentini's VoiceBank-DEMAND, respectively (denoted as "Model AA" and "Model VD", respectively, in Table S7), and use them to denoise our collected real-world recordings. For the denoised real-world audios, we measure the noise level reduction in detected silent intervals. This measurement is doable, since it requires no knowledge of noise-free ground-truth audios. As shown in Table S7, in terms of noise reduction, the model trained with our own AA dataset outperforms the one trained with the public VoiceBank-DEMAND dataset by a significant amount for *all* tested real-world recordings. On average, it produces **22.3 dB** noise reduction in comparison to **12.6 dB**, suggesting that our dataset allows the denoising model to better generalize to many real-world scenarios.

Table S7: Real-world recording noise reduction comparison.

| Real-world Recording | Model AA noise reduction (dB) | Model VD noise reduction (dB) |
|---|---|---|
| Song Excerpt 1 | **22.38** | 9.22 |
| Song Excerpt 2 | **21.16** | 17.65 |
| Chinese | **18.99** | 15.65 |
| Japanese | **16.95** | 9.39 |
| Korean | **25.90** | 9.76 |
| German | **14.54** | 7.29 |
| French | **21.95** | 17.16 |
| Spanish 1 | **33.34** | 11.00 |
| Spanish 2 | **31.66** | 14.09 |
| Female 1 | **17.64** | 7.79 |
| Female 2 | **30.21** | 8.64 |
| Female 3 | **19.15** | 6.70 |
| Male 1 | **24.81** | 14.44 |
| Male 2 | **24.92** | 13.35 |
| Male 3 | **13.10** | 10.99 |
| Multi-person 1 | **32.00** | 21.60 |
| Multi-person 2 | **15.10** | 15.07 |
| Multi-person 3 | **18.71** | 10.68 |
| Street Interview | **20.54** | 18.94 |
| AVERAGE | **22.27** | 12.60 |