[Reviews · NeurIPS 2020]

Review 1

Summary and Contributions: The paper presents an end-to-end architecture for speech denoising in which silence detection, noise estimation, and noise removal are jointly trained using a common objective function. Experiments are performed on data with noise synthetically added, using publicly available clean-speech and noise data. Comparisons are performed with several previously published models as well as a high quality commercial approach.

Strengths: The main novelty of the paper the proposal of an end-to-end model which implicitly learns non-speech segments from which the noise can be estimated. This gives a deep learning analog to spectral subtraction which can take account of non-stationary noise, which is very interesting. There is a set of experiments on the AVSPEECH corpus with noise added from DEMAND and AudioSet corpora, with good comparisons to other techniques. There are some good additional experiments, for example making use of ground truth speech/non-speech information to explore the effect of implicitly learning silence segments.

Weaknesses: The method relies on using mixed clean speech with noise for training, since ground truth information is required for the objective function. This in itself is OK, but there is no evaluation on "real" noisy speech data, which often results in somewhat different behaviour to constructed data. Indeed, as I understand it, separate models are constructed for each of the noise sets, so evaluations are carried out on mixes of the same types oif speech and noise data as used for training. Thus one does not know how dataset-dependent the method is - this appears to be particularly important for this method which is based on non-speech detection. I realise that much of the literature on speech denoising using artificially mixed data, and I realise that many of the objective evaluation methods rely on the availability of clean ground-truth speech, but this does not remove the problem. One could objectively evaluate on artificial test data using different clean speech and noise sources, which would tell us how the method copes with that kind of training/test mismatch. And one could perform evaluation using real recordings of noisy speech, for which clean ground truth does not exist, by using extrinsic evaluation approaches, for example those used for speech intelligibility. As it is, although the results look impressive, it is hard to know how they generalise either across datasets. (There are examples using speech in different languages but no evaluation.)

Correctness: The model description and experimental methodology are fine, given the above comments. It would have been informative to have also run on identical setups to some previous papers (e.g. [84]) so that comparisons could be made with published results rather than a retraining of the comparative approaches (it may be the case, for example, that there has been an implicit optimisation of hyperparameters for your approach on the data used, whereas this may not have been done for the approaches with which you compared).

Clarity: The paper is well-written - clear and I found it to be a good read.

Relation to Prior Work: Yes there is a very good discussion of previous work, and not just recently published work!

Reproducibility: Yes

Additional Feedback: UPDATE 2020-08-20: I have read the reviews and the author feedback, and I believe that the authors have made a good effort to engage with the reviews. My two main criticisms were to do with (1) experiments on real world date, and (2) cross-dataset experiments. For (1), I'm still not really convinced by the real world testing which is more of a demonstration than a test. I'm pleased that the authors can assert that they did not cherry-pick the data. However for (2), the authors have performed the basic cross-dataset experiments, and the results are only a small degradation. I view this positively. I also note the good effort to compare with some recently published papers. Although the lack of real-world experimentation prevents me from giving a strong accept, I am now happy to change my review score from 'marginal reject' to 'marginal accept' (assuming it is permissable for the paper to be updated with these new experimental results).


Review 2

Summary and Contributions: This paper presents a deep learning model for the speech denoising task by first modeling the silent intervals in the foreground which allows them to estimate the background noise and learn how it varies over time. It uses the key aspect in the spoken form of languages which makes human pause at various intervals in speech. The model uses both cleaned speech and noisy input coming from a single channel and doesn't use any other input modalities like videos or images. >>>> I thank the authors for their detailed response to the comments. The clarifications and experiments certainly improve the submission quality. Thanks for mentioning that the real-world examples were not cherry picked. One way for showing improvements on real-world dataset could have been to see if after applying the proposed model on noisy speech does the performance of downstream applications improve like ASR. I also acknowledge that I had missed some of the weaknesses that other reviewers mentioned. So overall I would like to keep my scores unchanged. <<<<

Strengths: 1) It uses an elegant approach to identify noisy patterns over time which is inherent in the spoken form of any human language. 2) The paper has done a really good job in identifying and comparing with previous work. 3) The paper has clearly presented various examples in which the model works / fails with actual audio as part of the supplementary material. 4) The model relies on single channel input which means that the approach can be applied to wide range of scenarios. 5) The model doesn't use any new input modality like videos and images which means that this approach will be complementary in presence of such modalities.

Weaknesses: 1) The method proposed in the paper is limited to the domain of speech denoising. 2) The model relies on silent interval detection in order to work at it's full potential. However, the paper clearly mentions this and shows that their model can also work without ground truth silent intervals.

Correctness: My major reservation for the correctness of this work is that the evaluation has been done on a synthetically produced noisy speech. A real noisy data is much more complicated because of unknown room acoustics, reverberations and distance of the noisy source from the microphone. One example of this dataset is the CHiME-3 dataset.

Clarity: The paper is well written, some evaluation about the real world cases would be good to mention it in the main paper instead of just keeping it in the supplementary material.

Relation to Prior Work: Excellent work with the relation to prior work.

Reproducibility: Yes

Additional Feedback: The experiment using Chinese, Japanese and Korean was mentioned only in the broader section. It would be better to mention them in the main paper (before the conclusion), specially since it's mentioned in the abstract.


Review 3

Summary and Contributions: This paper presents a new deep learning model for speech denoising. The proposed method was inspired by spectral subtraction methods and is based on detecting the silence regions, estimating the noise, and generating a mask to remove it. The authors compared the proposed system against several baseline models using the AVSpeech dataset with two noise datasets (Demand and AudioSet). Additionally, the authors evaluated the silence detection part of their model and conducted an ablation study over the loss function's different components.

Strengths: Overall, the paper is clearly written, it is easy to follow, and the main idea is well presented. The proposed method achieves impressive results, and the audio samples sound promising.

Weaknesses: Discussed below

Correctness: Yes

Clarity: Yes

Relation to Prior Work: Yes

Reproducibility: Yes

Additional Feedback: AFTER REBUTTAL: I would like to thank the authors for providing the additional experiments. I highly encourage the authors fo include them in the final version, it will make the paper and the general message stronger. Regarding real-world experiments, I advise the authors to launch a subjective study (MOS / ITU-T P.835 / ITU-T P.808) on real-world data to better evaluate the proposed method. Lastly, I suggest to clarify the VAD setup, I still think the results in Table 1 are a bit misleading as presented now. Adding a comment about it, will make it much better. ------------------------------------------------------------------------------------------------ Speech denoising is an essential task for various applications, and it should also be explored in the major ML conferences. However, some major issues need to be addressed. 1. There are plenty of research papers about speech denoising, and recently, several deep learning methods were also applied to the task. I do not expect the authors to compare to all of them, but there are several benchmarks for speech denoising/enhancement, and I would expect the authors to at least report results using one of them. Without such a comparison, it is very hard to understand the contribution / actual performance of the proposed method. Additionally, it makes it hard to reproduce the results. Such datasets include: valentini, DNS, MS-SNDS, etc. [1-4]. 1. There are new SOTA models for speech denoising. If the main contribution of this paper is to provide new SOTA, then the authors should compare it to them too. 2. It would be very helpful if the authors can also provide the same metrics for the noisy signal too. These results better highlight the relative improvement of each of the models and also the difficulty of the dataset. 3. I find the comparison in Table 1 misleading and unfair. The Google WebRTC VAD was optimized to detect activity using such SNRs. Additionally, the proposed model was trained on the same conditions! While the other baselines not. Minor comments: 1. line 4: "...there is often a short pause between each sentence or word..." This might be true for sentences, but not for words. Word discovery is actually a very challenging task. I suggest rephrasing this sentence. 2. line 115: you mentioned state-of-the-art methods, however the authors do to compare to SOTA methods. 3. line 132: 1/30. It would be better to report ms instead of 1/30 second. 4. line 139: sentence not clear [1] Fu, Szu-Wei, et al. "MetricGAN: Generative adversarial networks based black-box metric scores optimization for speech enhancement." arXiv preprint arXiv:1905.04874 (2019). [2] https://datashare.is.ed.ac.uk/handle/10283/2791 [3] https://github.com/microsoft/DNS-Challenge [4] Reddy, Chandan KA, et al. "A scalable noisy speech dataset and online subjective test framework." arXiv preprint arXiv:1909.08050 (2019).


Review 4

Summary and Contributions: The paper decomposes the popular denoising technique called "mapping approach" into 3 subproblems. First is akin to doing VAD, second is explicit noise spectrum estimation, third is usual denoising. Through ablation study, they express that their novel silence detection module works. Enhancement metrics are reported using 2 popular noise corpora Audioset and DEMAND and comparison is also done using existing method codes.

Strengths: Silence detector (first module of their proposed network) looks promising to learn VAD. Complex mask (and complex signal representation) are used which is experimented less in the field.

Weaknesses: While the methodology presented in this work is interesting, there exists several new research works which do very well on DEMAND. Comparison/mention of them is missing Examples: 1. End-to-End Multi-Task Denoising for joint SDR and PESQ Optimization 2. T-GSA: Transformer with Gaussian-Weighted Self-Attention for Speech Enhancement 3. Speech enhancement using self-adaptation and multi-head self-attention 4. Deep Residual-Dense Lattice Network for Speech Enhancement

Correctness: Technically, it is correct.

Clarity: Yes, clear presentation.

Relation to Prior Work: It misses comparison/mention with/of several recent SOTA methods.

Reproducibility: Yes

Additional Feedback: Using standard training data and comparison with existing techniques is desirable. [after rebuttal] I have increased my score from 5 to 6 *in strong hope* that results on VCTK/DEMAND dataset will be included in final version.

[Author Response · NeurIPS 2020]

1. Dataset and SOTA benchmark comparisons [R1, R3, R4]: Several of the SOTA methods mentioned by the reviewers are concurrent to our work, appearing a few weeks before NeurIPS deadline. During the rebuttal period, we performed benchmark comparisons to these methods (suggested by R3 and R4), including TF-GAN (Soni et al. 2018), Metric-GAN (Fu et al. 2019), SDR-PESQ (Kim et al. 2019), T-GSA (Kim et al. 2020), Self-adaptation DNN (Koizumi et al. 2020), and RDL-Net (Nikzad et al. 2020). Trained on the same dataset (i.e., Valentini's DEMAND) as in those methods and tested in the same way, our method is highly competitive to the best of those methods under each metric (different existing methods perform the best under different metrics). For example, under PESQ, our method produces **3.16**, while existing methods reported the following scores: 2.53 (TF-GAN), 2.86 (Metric-GAN), 3.01 (SDR-PESQ), 3.06 (T-GSA), 2.15 (Self-Adaptation DNN), 3.02 (RDL-Net). Under CBAK, our method produces **3.54**, while others are: 3.12 (TF-GAN), 3.18 (Metric-GAN), 3.54 (SDR-PESQ), 3.59 (T-GSA), 2.82 (Self-Adaptation DNN), 3.43 (RDL-Net). We also evaluated speech intelligibility using STOI. Our method produces **0.98**, while RDL-Net reported 0.94 (other methods do not report this score). We are happy to include a full set of comparison scores in the revision of the paper.

**Why creating our own dataset?** We would also like to clarify that there are strong reasons for creating our own dataset. First, it is important to train/test denoising models under a wide range of noise levels, including strong noise (i.e., low SNRs). Unfortunately, existing datasets have limited ranges of SNR levels. For example, DEMAND consists of audios with SNR in [0dB, 15dB]. This is significantly narrower than what we wish to test against (e.g., with an SNR of -10dB), because many real-world recordings, as we show in supplementary material, have strong noise (much lower than 0 dB SNR). Secondly, a key challenge faced by any denosing model is to suppress *structured* noise. Existing datasets like DEMAND provide several kinds of environmental noise, but lack the richness of other types of structured noise (such as music). The lack of noise diversity makes the datasets less ideal for denoising real-world recordings.

To support our rationale, we performed an experiment to compare our dataset (AVSPEECH+Audioset) with DEMAND for real-world denoising. We train two versions of our model using our dataset and DEMAND, respectively, and use them to denoise real-world recordings (those shown in supplementary material). For the denoised real-world audios, we measure the noise level reduction in detected silent intervals. This measurement is doable, since it requires no knowledge of noise-free ground-truth audios (which are not available for real-world recordings). In every tested real-world recordings, the noise reduction from the model trained with our dataset is significantly higher than that from the model with DEMAND. On average, it produces **22.3 dB** noise reduction in comparison to **12.6 dB**, suggesting that our dataset allows the denoising model to better generalize to many real-world scenarios.

2. Test on real-world data [R1, R2, R3]: We believe that all methods must be examined against real-world data. Quantitative evaluation on real-world data, however, is not easy. To evaluate a denoised signal, all the metrics (like SSNR, PESQ, CBAK, STOI, etc.) require to know the clean, ground-truth signal, which is not available for real-world recordings. It is for this reason that almost all existing denoising models were evaluated on synthetic data, and we follow this common approach to evaluate our model quantitatively.

Nevertheless, we did one step further than previous works. In the supplementary webpage, we reported denoising results of various *real-world recordings*. These are recordings we downloaded online or recorded in daily environments. These are recordings in diverse scenarios: in a driving car, a café, a park, on the street, in multiple languages (Chinese, Japanese, Korean, German, French, etc.), with different genders and accents, and with singing songs. None of these recordings is cherry picked. And we also showed the results from other methods, including the widely used professional audio processing software, Adobe Audition. The point we wish to convey is: testing our method with real-world data and letting the reader to judge by themselves how well our method can generalize to different real-world scenarios. While these are qualitative results, to our knowledge no previous work has reported such diverse real-world results.

3. Generalization across datasets [R1, R3]: Our real-world results in the supplementary material serve for demonstrating the generalization ability of our model in various real-world scenarios. In addition, we performed *three* cross-dataset tests to evaluate the generalization ability of our model, as suggested by R1. **i)** We train our model on Audioset but evaluate it on DEMAND testset—the same testset as in other methods. The PESQ score is **3.00**. As comparisons, our method trained on DEMAND training set yield a PESQ score of 3.16, and the best score reported in previous methods (trained and tested on DEMAND) is 3.06 (T-GSA). **ii)** We train our model on AVSPEECH+Audioset and evaluate it on our second dataset in which the noise is from DEMAND. The PESQ score is **2.65**, and for comparison, our model trained and tested with DEMAND noise distribution yields a core of 2.80. **iii)** We train our model on our second dataset (in which the noise is from DEMAND) and test it on AVSPEECH+Audioset. The PESQ score is **2.12** in comparison to 2.30 from our model trained and tested on Audioset noise distribution. We also note that we couldn't compare the generalization ability of our model with existing methods, as no previous work reported cross-dataset evaluation results.

4. Metrics for the noisy signal [R3]: Thanks for your suggestion. We measured the noisy signals under the six metrics we used. Corresponding to what we reported in Fig. 5, these scores are: 2.72 (PESQ), 3.04 (SSNR), 0.85 (STOI), 3.55 (CSIG), 2.70 (CBAK), 3.05 (COVL) for DEMAND noise, and 1.91, 3.46, 0.71, 2.79, 2.34, 2.27 for Audioset.

5. Google WebRTC VAD [R3]: We understand that VAD is designed to work with low-noise signals. It is included in Table 1 merely as an alternative approach that can detect silent intervals. We will clarify this point in the revision.

[Meta-Review · NeurIPS 2020]

While novelty is on the weak side for NeurIPS, reviewers found it overall simple and elegant. Results on synthetic data are found impressive, and proposed audio clips seem promising. Authors addressed most of the reviewers concerns with their rebuttal. The only major point remaining is how one could benchmark on real-world data. Two possible paths were suggested by reviewers during the discussion: i) use the DNS dataset (but this one was published only 4 months ago) ii) show improvement on a downstream task like ASR - but this would require quite a bit of work and could be viewed as been out-of-scope for this paper. The consensus was thus overall leaning towards acceptance.